# The Kinetics of Amyloid Fibril Formation by *de Novo* Protein Albebetin and Its Mutant Variants

**DOI:** 10.3390/biom10020241

**Published:** 2020-02-05

**Authors:** Vitalii Balobanov, Rita Chertkova, Anna Egorova, Dmitry Dolgikh, Valentina Bychkova, Mikhail Kirpichnikov

**Affiliations:** 1Institute of Protein Research, Pushchino, Moscow 142290, Russiabychkova@vega.protres.ru (V.B.); 2Shemyakin-Ovchinnikov Institute of Bioorganic Chemistry, Russian Academy of Sciences, Miklukho–Maklaya st. 16/10, Moscow 117997, Russia; 3Faculty of Biology and Biotechnologies, National Research University Higher School of Economics, Moscow 117312, Russia; 4Biology Department, Lomonosov Moscow State University, Leninskie gory, 1/12, Moscow 119899, Russia

**Keywords:** amyloidogenesis, amyloid fibrils, cross-beta structure, albebetin, *de novo* proteins

## Abstract

Engineering of amyloid structures is one of the new perspective areas of protein engineering. Studying the process of amyloid formation can help find ways to manage it in the interests of medicine and biotechnology. One of the promising candidates for the structural basis of artificial functional amyloid fibrils is albebetin (ABB), an artificial protein engineered under the leadership of O.B. Ptitsyn. Various aspects of the amyloid formation of this protein and some methods for controlling this process are investigated in this paper. Four stages of amyloid fibrils formation by this protein from the first non-fibrillar aggregates to mature fibrils and large micron-sized complexes have been described in detail. Dependence of albebetin amyloids formation on external conditions and some mutations also have been described. The introduction of similar point mutations in the two structurally identical α-β-β motifs of ABB lead to different amiloidogenesis kinetics. The inhibitory effect of a disulfide bond and high pH on amyloid fibrils formation, that can be used to control this process, was shown. The results of this work are a good basis for the further design and use of ABB-based amyloid constructs.

## 1. Introduction

Protein misfolding and aggregation, when monomeric protein molecules turn into amyloid fibrils, is one of the factors causing severe disorders, including neurodegenerative diseases [1]. Amyloids are fibrils, usually unbranched, consisting of protein monomers linked by hydrogen bonds between β-strands of an intermolecular β-sheet; they lie perpendicularly to the axis of the fibril. This structure is known as the cross-beta structure and is considered a universal sign of amyloid fibrils [2,3].

It is important to mention that there is evidence of amyloid formation by proteins under conditions associated not only with pathology, but also with the implementation of the key biological functions of these proteins [4,5,6,7]. There are proteins exhibiting functional activity only while having an amyloid configuration [8,9,10]. Numerous facts demonstrate that amyloidogenesis is a fundamental property of the polypeptide chain, which allows one to consider the amyloid structures as variants of protein quaternary structure. The formation of amyloid structures is typically mediated by protein transformation due to certain factors to an intermediate state which properties are similar to those of a molten globule (i.e., the presence of a secondary structure but no dense packing of side groups) [11]. Such states, as a rule, have greater mobility of the polypeptide chain and increased hydrophobicity [12]. This, as many authors suppose, leads to increased aggregation ability compared to the native state. This comparison, however, is not applicable to natively unfolded and highly destabilized proteins, since they initially do not have a native state, which eliminates the need to cross the free-energy barrier between the native and intermediate states.

Albebetin (ABB), a *de novo* designed protein with a predesigned spatial structure the topology of which is not found in natural proteins, has the following properties [13]. Albebetin is an ~8 kDa polypeptide chain consisting of 73 amino acid residue (AAR). It has a predesigned secondary structure (two identical α-β-β repeat motifs), but does not exhibit dense packing of side groups and exists in the state close to that of a molten globule [13]. The absence of tight packaging makes structural and thermodynamic studies of this protein difficult. For the same reason, it is difficult to use computational methods for calculating the mobility of the protein chain and the solvent accessible surface area. In our work, we will use the theoretically predicted structures.

Albebetin was used in our previous works to illustrate the fundamental possibility to utilize *de novo* designed proteins as efficient carriers of biologically active peptide sequences. Thus, an artificial protein albeferon [14], which is structurally identical to albebetin with the inserted active fragment of human interferon α_2_ (IFN-α_2_), retained the predesigned structure of the original albebetin [15], exhibited high biological activity (comparable to that of IFN-α_2_) [16], and low immunogenicity [17]. Next, a number of artificial proteins were designed based on albebetin and albeferon; these proteins carried not only functional peptide fragments of natural IFN-α_2_ [18], but also of differentiation factor HLDF [19] and of human insulin, exhibiting the respective biological activities and relatively low immunogenicity [20,21].

Earlier, we demonstrated that the *de novo* protein ABB and its derivatives carrying biologically active fragments of natural proteins can form fibrils at physiological pH [22,23,24]. The amyloid nature of these aggregates was verified by analyzing far-UV circular dichroism (CD) spectra and the aggregate morphology by atomic force microscopy (AFM). It was shown that addition of short peptides (the octapeptide of human INF-α_2_ or the hexapeptide HLDF-6 of the differentiation factor of human promyelocytic leukemia cell line HL-60 (HLDF)) to the N-terminal end of ABB significantly enhances the amyloidogenic properties of the protein and increases the rate of amyloid fibrils formation. A similar effect was also observed in the presence of the free hexapeptide HLDF-6 in ABB solution [22]. Higher ionic strength of the solution also accelerates the formation of ABB amyloids, although to a lesser extent than adding the peptides [23]. Hence, the artificial protein ABB and its modified variants can be used as a convenient model to study the mechanisms of fibril formation and the cross-beta core structure of amyloid fibrils.

Mutations are often the significant reason of why a normally functioning protein becomes prone to amyloid fibril formation [1]. In this connection, it would be interesting to model such amino acid substitutions in the ABB that could change the kinetics of amyloid formation in either side. For this reason, a thorough study of the amyloidogenic properties of original ABB and its mutant variants carrying substitutions that enhance/attenuate the amyloid formation ability of the protein is relevant, as well as investigating the influence of external factors on this process. These studies can shed light on the molecular mechanisms underlying amyloidogenesis, which in the long run will make it possible to use amyloid structures to design novel biomaterials having a broad scope of application, as well as potential therapeutic agents capable of inhibiting or accelerating the amyloid formation.

In this study, the amyloid fibril formation by the *de novo* protein ABB is scrutinized using a number of spectral and microscopic methods. The effect of some point mutations on this process is evaluated. In addition, we discuss the prospects for the use of ABB as an amyloid carrier of functional domains and the possibility of using the detected effects to control amyloid formation in practical applications.

## 2. Materials and Methods

### 2.1. Constructing Mutant ABB Genes

The mutant genes of artificial protein ABB were obtained using a PCR series. The protein ABB consists of two identical fragments of amino acid sequence. It is clear that the two perfect repeats in the ABB sequence impede the introduction of mutations by PCR. We suggested using a two-stage process to obtain the mutant ABB genes.

At the first stage, two PCR reactions were performed as follows: the 5’-end and 3’-modified primers were employed for the one PCR experiment, while the 5’-modified and 3’-end primers were used for another PCR experiment. Each reaction yielded two products, since the mutant primers bind to the gene regions carrying codons for both mutations (His30 and His65). The amplification reactions were conducted using the ABB gene within a pET-32 LIC plasmid vector (Novagen, Kenilworth, NJ, USA) as a template. The DNA fragments obtained in these PCR experiments, which presumably carried the desired mutations and whose molecular weight matched the calculated one, were used to produce full-length fragments carrying the desired mutations by PCR-2.

At the second stage, the products of PCR-1 and PCR-2 reactions carrying a H30F point substitution were used for PCR-3, and the products carrying a H65F point substitution were employed for PCR-4. The DNA fragments carrying the mutations were completed, and the resulting full-length fragments with H30F and H65F substitutions were used as templates in the next amplification cycles. Since we have demonstrated that the disulfide bond plays a significant role in amyloid formation, it was interesting to synthesize and study the albebetin variant carrying a double mutation at Cys residues. The gene of the mutant variant ABB(C8S/C43S) was constructed and synthesized according to the scheme described above. The nucleotide sequences of mutant genes in the plasmid DNA were determined on an ABI Prism 3100-Avant Genetic Analyzer (Applied Biosystems, Waltham, MA, USA).

Full-length DNA fragments were cloned into the plasmid vector pET-32-ABB LIC via the ligation reaction. The reaction medium (10 µL) contained the following components: buffer for T4 DNA ligase (Fermentas, Vilnius Lithuania), T4 DNA ligase, an insert—gene of ABB or its mutant, and plasmid vector pET-32 LIC/*Bam*H I/*Hin*d III. The reaction was carried out at 16 °C during 1 h.

### 2.2. Gene Expression; Isolation and Purification of Recombinant Proteins

The plasmid vector pET-32-ABB LIC allows one to express the target protein gene fused with the *trx* gene of thioredoxin within a hybrid protein (TRX-ABB). The hybrid protein also carries the 6×His sequence to ensure rapid and efficient purification of the product using metal chelate chromatography. The expression level of hybrid protein in this construct was appreciably high, making it possible to quantitatively produce the target proteins. The expression of mutant ABB genes was performed in *E. coli* strain BL-21 DE3. The cells transformed with the plasmid carrying the gene encoding the synthetic protein were cultured in Terrific Broth liquid growth medium supplemented with ampicillin as in [18,19] to a final concentration of 100 µg/mL at 37 °C under vigorous stirring (250 rpm) for 22–24 h. Once expression had been completed, *E. coli* cells were precipitated by centrifugation at 6000 g and 4 °C for 15 min.

The resulting cell biomass was resuspended in 50 mM Tris-HCl, pH 8.0, 1 mM EDTA, 0.02 M β-mercaptoethanol; MgCl_2_ (to a final concentration of 5 mM) and DNase I (Sigma-Aldrich, St. Louis, MO, USA) (50 µg per g of cells) were added. The cell suspension was homogenized on a French press (Spectronic Instruments, Inc., Irvine, CA, USA). The water-insoluble fraction of the homogenate was precipitated by centrifugation at 30,000 rpm for 30 min; approximately 90% of hybrid protein was present in the water-soluble fraction. The hybrid protein was isolated from the cell lysate by ion-exchange chromatography on a DEAE-cellulose column (Sigma-Aldrich, St. Louis, MO, USA) in 20 mM Tris-HCl, pH 8.0, 100 mM NaCl. The hybrid protein was eluted with 50–500 mM NaCl gradient in the same buffer as in [18,19].

Next, the hybrid protein was purified on a column packed with nickel-nitrilotriacetic acid (Ni-NTA) metal chelate resin (Qiagen, Hilden, Germany) equilibrated with 50 mM Tris-HCl buffer, pH 8.0, containing 300 mM NaCl. The hybrid protein was eluted with 20–250 mM imidazole gradient. The purified hybrid protein was concentrated on the DEAE-cellulose column until a concentration of at least 10 mg/mL. The hybrid protein was subjected to enzymatic cleavage using factor Xa protease (Sigma-Aldrich, St. Louis, MO, USA) in 20 mM Tris-HCl buffer, pH 8.0, containing 100 mM NaCl and 5 mM CaCl_2_, at room temperature for 24 h at a 1:2000 molar ratio between the enzyme and the protein. The reaction mixture was applied to the column packed with Ni-NTA in the respective buffer; the target albebetin was not bound to the sorbent and was eluted from the column immediately after the void volume mark as in [18,19].

### 2.3. Protein Purity Analisis

All stages of protein isolation and purification were controlled by denaturing gel electrophoresis in 12% Tris-Tricine PAGE gel with 1% SDS [25] and tris(2-carboxyethyl)phosphin hydrochloride (TCEP), a disulfide bound reducing agent. Gel electrophoresis under nondenaturing conditions in 9% PAGE gel was used to evaluate the homogeneity of the resulting protein under native conditions and to verify whether there were any aggregates in the solution.

### 2.4. Absorption Spectroscopy

Absorption spectra were recorded on a Cary-100Bio spectrophotometer (Varian, Palo Alto, CA, USA). Protein concentrations were determined using the calculated extinction coefficients A^1 cm^_1 mg/mL_ = 15 at λ = 220 nm for ABB and its mutant and A^1 cm^_1 mg/mL_ = 0.6 at λ = 280 nm for hybrid proteins.

### 2.5. Fluorescence Spectroscopy

Thioflavin T (ThT) (Sigma, St. Louis, MO, USA) was used as a cross-beta specific fluorescent dye. The final concentration of Thioflavin T was 0.1 µM. The fluorescence measurements were made on a Cary Eclipse fluorescence spectrophotometer (Varian, Palo Alto, CA, USA) in quartz cells 3 × 3 mm. Continuous monitoring was performed at an excitation wavelength of 455 nm; the emission was recorded at 480 nm. When collecting the samples after certain time intervals, the fluorescence spectra were recorded at 448–600 nm at an excitation wavelength of 455 nm. 

When studying the kinetics of amyloid aggregation of hybrid protein, the intrinsic fluorescence of tryptophan residues was also recorded at 300–450 nm, at an excitation wavelength of 280 nm.

### 2.6. Light Scattering

The changes of light scattering intensity during aggregate growth were recorded by continuous monitoring in a cell. Light scattering was measured simultaneously with recording the fluorescence intensity of ThT for the same samples at an excitation wavelength of 455 nm; the recording wavelength was 449 nm.

### 2.7. CD Spectroscopy

The far-ultraviolet (UV) CD spectra were recorded on a Chiroscan spectropolarimeter (Applied Photophysics Ltd, London, UK) in a cell with an optical path length of 3 mm in the wavelength range of 190–250 nm. Molar ellipticity [Θ] was calculated taking into account the concentration of the protein, its amino acid composition and the dimension of the cuvette [26].

### 2.8. Electron Microscopy

To prepare the specimens, the protein solution after the incubation was diluted to 0.1 mg/mL. The specimens were adsorbed onto a carbon film and subjected to negative staining with 1% aqueous solution of uranyl acetate. Electron microimages under magnification 220,000 and 160,000 were recorded on a JEM100C electron microscope (JEOL, Tokyo, Japan).

### 2.9. Atomic Force Microscopy

A solution of the incubated specimen was diluted to a concentration of 0.1 mg/mL. A specimen droplet containing the aggregates was applied onto the cleaved mica surface. After 5-min incubation, the specimen was washed with water and allowed to dry at room temperature. Silicon probes with the tip curvature radius of ~10 nm (NT-MDT, Moscow, Russia) were used for the semi-contact mode. Their resonance frequency was varied for different probes within 60–120 kHz. The specimen was scanned at a resolution of 256 × 256 or 512 × 512 points. 

### 2.10. Confocal Fluorescence Microscopy

The size and morphology of the resulting aggregates were evaluated using a Leica TCS SPE confocal fluorescent microscope. After incubation, the protein solution was diluted with Prolong Gold anti-fade solution (Invitrogen, Waltham, MA, USA) at a 1:1 ratio. The specimens (5 µL) were applied onto the microscope slides and covered with a cover slip slide. Thioflavin dye fluorescence was laser-excited at 405 nm. Confocal fluorescence microscopy images were recorded in a wavelength range of 450–600 nm. Serial confocal images were obtained at a 0.5 µm scan step between the focal planes.

### 2.11. Error Calculation

Two types of errors were used in the work. The first is the error of the measuring instruments. This indicator is a characteristic of devices and was determined when baselines were measured. It was used to compare data obtained in the same experiment with different instruments.

The second type of error reflects the relative standard deviation from the mean value upon repeated repetition of the experiment. This error was taken into account when comparing data for various mutant forms of ABB or when assessing the impact of conditions.

## 3. Results and Discussion 

### 3.1. Studying the Artificial Protein ABB Amyloid Formation 

The experimental investigation of amyloid formation kinetics of remains a nontrivial problem. Amyloid formation by proteins depends on a number of stochastic factors, which often cannot be taken into account [27]. As a result, there are significant dispersions in the process rate and observed signal amplitude when the experiments are repeated. The aforementioned dispersion makes it impossible to compare the data obtained using different methods in different experimental runs. In order to solve this problem, we simultaneously conducted the measurements using several methods at the one and the same sample. In this case, we can take into account only the errors of the instruments, which are incomparably smaller than deviation upon repetition of the experiment.

Protein solutions were incubated in a thermostated cell at 45 °C in a 20 mM Tris-HCl buffer, pH 7.4, containing 200 mM NaCl, at protein concentration of 10 mg/mL, in the presence of tris(2-carboxyethyl)phosphin (TCEP), a disulfide bonds reducing agent. The thioflavin T (ThT) dye fluorescence and the intensity of scattered light were continuously measured during incubation. Samples were collected from the cell after certain intervals to record the circular dichroism spectra and conduct other measurements. The morphology of the aggregates was studied by fluorescence microscopy, electron microscopy, and atomic force microscopy. The monomer content and the amount of nascent aggregates were estimated by electrophoresis under nondenaturing conditions.

#### 3.1.1. Fluorescence Spectroscopy and Light Scattering

The kinetics of amyloid formation has been studied by continuously recording the fluorescence intensity of ThT dye added to the protein solution. In order to evaluate the rate at which the size of nascent amyloids increased, the light scattering spectra were recorded simultaneously with the fluorescence (Figure 1). The results demonstrate that the intensity of ThT fluorescence increased upon incubation of the sample due to the emergence of the cross-beta structure. The intensity of scattered light was simultaneously increased, indicating that larger aggregates were formed. The light scattering and fluorescence curves are S-shaped. The slight difference in the curve shape is possibly related to aggregate formation and accumulation, followed by their conversion to the form that specifically binds ThT. A short, relatively to the time of growth, lag-period (~10 min vs. ~150 min) testifies [28] a linear (without branching or fragmentation) growth of the amyloid and a relatively large (~10 monomers) size of the fibril nucleus. It is important to note that upon repetition a sufficiently large distribution mentioned above is found.

#### 3.1.2. Far-UV Circular Dichroism

Amyloid formation is accompanied by significant changes in the β-structure content. This makes it possible to use CD spectroscopy to study the amyloidogenesis processes. The Far-UV CD spectra of the specimens were recorded prior to incubation, as well as at 20 and 270 min after the beginning of the incubation (Figure 2b). The shape of the CD spectra demonstrates that the secondary structure of the ABB protein has changed severely.

One can easily see that the most significant changes take place at wavelengths of 220 and 205 nm. The time dependences of the ellipticity Θ at 205 and 220 nm allowed us to infer that the negative ellipticity was initially slightly increased in both cases and then decreased. The time dependence of the ratio of the ellipticities at these wavelengths (Figure 2a) allows one to estimate the changes in the shape of the CD spectra over time.

An analysis of the spectral data infers that the shape of the CD spectra is significantly altered in the beginning of aggregation (related to the increasing ellipticity at 205 nm); however, there are only minor changes in the intensity of ThT fluorescence. This fact indicates that intermediate aggregates the secondary structure of which differs from that of the monomeric form of the protein have accumulated, but have not bound ThT yet. Further changes observed using these two methods are also asynchronous: the shape of the CD spectra changes at a slower rate than the rate at which the intensity of ThT fluorescence increases. This behavior demonstrates that intermediate structures (protofibrils) have been accumulated: they have already bound the dye ThT, but their secondary structure does not match that of the final aggregates yet. In all likelihood, fibril maturation and formation of the final secondary structure occur at the final stage.

#### 3.1.3. Electron Microscopy

Electron microscopy was employed to characterize the size and to study the morphology of the resulting aggregates. Electron microimages of ABB were recorded after incubating it at 45 °C for 10, 30, and 300 min (Figure 3).

One can see in the images that small-sized globular aggregates are detected at the initial stage of amyloid formation (Figure 3a). Next, extended fibrils are formed (Figure 3b). Their size increases over time, and a fibril network is eventually formed (Figure 3c).

#### 3.1.4. Atomic Force Microscopy

According to the AFM data, ABB forms conventional amyloid fibrils in addition to small-sized globular aggregates. The kinetics of aggregate morphology changes has been studied by AFM. Figure 4 shows the images of amyloid structures formed by ABB observed after different incubation periods. Again, it is seen that small aggregates are formed at short incubation periods (5–30 min), followed by formation of extended fibrils (at 1–2 h), which eventually grow larger to form a fibril network (at 3–6 h). After incubation for 6 h, the fibril network becomes very dense. Both individual fibrils and bundles consisting of several thin fibrils can be detected in it.

#### 3.1.5. Confocal Fluorescence Microscopy

Fluorescence microscopy has a lower resolution compared to AFM and electron microscopy. Although it does not see oligomers at all, it allows one to examine larger fields and larger structures. Furthermore, this examination technique does not involve sample sorption onto the surface, which reduces the risk of artifacts. The kinetics of amyloid formation by ABB observed by fluorescence microscopy is shown in Figure 5. One can see that fibrils start aggregating into clusters after incubation for 30 min. The size of these clusters then increases, and large bundles are formed. After 6 h incubation, these bundles become as long as several hundred micrometers. Detailed examination of the bundles reveals a longitudinal packing of fibrils. This means that fibrils are oriented along the major axis in this bundle.

### 3.2. The Effect of External Factors on Amyloid Formation

Such external factors as pH and the presence of agents breaking disulfide bonds were found to be most significant.

Variation in pH within the range from pH 7 to pH 5 (Figure 6a) had the most substantial influence on amyloid formation, whose efficiency was the highest at pH 5.5 (this corresponds to the titration of side groups of His residues). In an ABB molecule, these residues are located at positions 30 and 65 in identical structural elements (namely, in the internal β-strands of ABB molecule) and are solvent-exposed (Figure 7). It was found by analyzing the amino acid sequence of albebetin that introduction of mutations at these residues (substitution of His residue for the uncharged Phe residue) may give rise to regions characterized by high expected contact density [29] in addition to changes in charge distribution, which would contribute to the formation of additional amyloidogenic domains.

The disulfide bond in albebetin can be formed between Cys residues at positions 8 and 43, which are symmetrically located in identical structural elements of the ABB molecule (as it is in the case for His30 and His65 residues). This bond`s formation increases the content of the secondary ABB structure (see Figure 6b). We conducted experiments to study kinetics of amyloid formation by ABB in the presence of TCEP, a disulfide bonds reducing agent (Figure 6b). Furthermore, it was demonstrated that in the absence of the reducing agent TCEP, the protein containing the disulfide bond forms amyloid fibrils at a much lower rate (several weeks rather than days; data not shown), which is consistent with the findings obtained using non-reduced ABB protein [22,23]. The results of our study may indicate that the conformational mobility of the polypeptide chain is very important for efficient amyloid formation and the disulfide bond plays a crucial role as one of the factors stabilizing the non-amyloid form of the protein. The removal of one of three disulfide bonds in an insulin molecule is known to destabilize the native state and enhance amyloidogenesis [31], while the insertion of the fourth non-canonical S-S bond between the Cys-substituted residues of chains A and B prevents the formation of amyloid structures [32].

However, it should be mentioned that the role of disulfide bond in amyloid fibril formation is not always interpreted using the paradigm of native state stabilization. Thus, it has been demonstrated for the highly amyloidogenic two-chain H-fragment of insulin that the presence of a disulfide bond plays a decisive role for its amyloidogenicity: association of H-fragment monomers to fibrils decreased abruptly in the presence of an agent reducing disulfide bonds [33]. In a similar manner, disulfide bond rupture induced by sodium tetrathionate inhibited amyloidogenesis of hen egg white lysozyme (HEWL) at pH 2.0 [34].

Hence, the role played by disulfide bond in amyloidogenesis is rather protein-specific; it can both stimulate and inhibit the amyloidogenesis. In the case of amyloid formation by albebetin protein, disulfide bond has a stimulating effect.

### 3.3. Design of Mutant Variants of ABB and Study of Their Amyloidogenicity

As mentioned above, the process of amyloid formation of ABB is sensitive to pH changes in the region of histidine residues titration. Such residues of ABB are located at positions 30 and 65. An analysis of the surface distribution of charged and polar residues in the predicted spatial ABB structure (Figure 7a) shows, residues in these positions are surrounded by glutamic acids, which have a negative charge in examined pH region. Titration of histidine residues leads to the positive charge appearance and a significant change in the surface charge distribution, which in turn contributes to amyloid formation. Since there are two identical repeats in ABB sequence, we decided to check whether they are identical in the process of amyloid formation. For this, two mutant variants were designed that carry H30F and H65F substitutions.

The second issue that we wanted to clarify with the help of mutational analysis is the effect of cysteine residues on the process of amyloid formation. As can be seen in Figure 7c, cysteine residues that did not form a disulfide bond are located on opposite sides of the protein. This leads to the fact that either intermolecular disulfide bonds are formed, or the protein structure is strongly deformed. This leads to heterogeneity in the albebetin population with oxidized cysteines. At the same time, for this entire population, a significantly reduced ability to amyloid formation is observed. To eliminate the heterogeneity of the population, we made mutations replacing existing cysteines and obtained a double mutant C8S/C43S.

The expected practical use of amyloid fibrils is associated with the creation of fibrils decorated with functionally active domains. The intermediate stage of isolation of ABB is a fusion protein with thioredoxin. In the course of present work it was noted that under certain conditions this construct also forms amyloid-like aggregates. It was decided to study this process in some more details. Great importance was the determination of the role of each of the parts of the fusion protein in amyloid formation. This did not require the creation of additional genetic constructs.

#### 3.3.1. Studying the Kinetics of Amyloid Formation by the Mutant Variants ABB(H30F) and ABB(H65F)

Are the same ABB repeats the same?

When studying amyloid formation by the mutant variant ABB(H30F), we demonstrated that the rates of increase intensity of ThT fluorescence and aggregate accumulation are slower than those for the original ABB (Figure 8a).

This fact is consistent with the results of electrophoretic analysis of protein specimens under non-denaturing conditions, which demonstrate that the aggregated form of both the original type and the mutant ABB proteins emerges during the first hour of incubation; however, accumulation of the aggregated form of ABB(H30F) protein is a slower process (Figure 8b,c). Identically to the case of ABB, the monomeric form is almost completely converted into the aggregated form after 32 h incubation. It is important to mention that the monomeric form of ABB(H30F) protein was retained even after the specimen had been incubated for more than 50 h. 

No significant differences were observed for the kinetics of amyloid formation by ABB(H65F) and the morphology of its amyloid structures (data not shown); they appeared to be largely similar to those of the original ABB (see Figure 1, Figure 2, and Figure 4). Substitution H30F reliably slows down the rate of amyloid formation compared to original ABB and ABB(H65F). Hence, the experimental data infer that the H65F and H30F substitutions are not equivalent for fibrillogenesis despite the fact that they are located in identical positions - two identical α-β-β protein repeats. The amyloid structures of all these proteins observed by AFM have similar morphologies and sizes (data not shown).

Thus, from the point of view of amyloid formation, the albebetin repeats are not identical, and this should be taken into account in the further design of structures based on it. The same can apply to many other proteins containing amino acid repeats.

#### 3.3.2. Studying the Kinetics of Amyloid Formation by the Mutant Variant ABB(C8S/C43S) 

According to the CD spectroscopy data, the structure of the mutant protein ABB(C8S/C43S) was similar to that of the original ABB treated with an agent reducing disulfide bonds (Figure 9a). As expected, the introduction of a C8S/C43S double mutation can enhance the amyloidogenic properties of ABB protein. In addition, these mutations can save us from the need to use agents breaking the S-S bond and modifying cysteine residues. This in turn would allow us to eliminate the uncertainties associated with improper tying of intra- or intermolecular S-S bonds.

However, this fact caused some difficulties when studying the kinetics of amyloid formation. In the experiments with original ABB, the instant of initiation of amyloid aggregation coincided with the instant when the agent reducing disulfide bonds was added. For the mutant variant ABB(C8S/C43S), the amyloid structures started forming at the specimen preparation stage (Figure 9b), while the increasing temperature and changing pH only accelerated this process. According to the AFM data, the morphology of the resulting fibrils was close to the one observed for the fibrils formed by original ABB (data not shown).

Thus, eliminating the ABB ability to form disulfide bonds, we deprive ourselves of an amyloid formation control tool. The use of such a mutation is justified only when the intended use of amyloid fibrils precludes the use of a reducing agent.

#### 3.3.3. Studying the Kinetics of Amyloid Formation by the ABB-TRX Hybrid Protein

As mentioned above, it was noted that under certain conditions, the TRX-ABB fusion protein also forms amyloid-like aggregates. This process has been investigated in more detail. It was demonstrated in this study that both ABB-TRX in the presence of TCEP and ABB(C8S/C43S)-TRX can form amyloid fibrils (Figure 10a). We determined that under the examined conditions, TRX with a linker, separately from ABB, does not form amyloid fibrils. This indicates that it is ABB that is the amyloidogenic part in the composition of the fusion protein. We see that the ABB has not lost its ability to form amyloid fibrils. This suggests that the N-end of ABB (by which it is connected to TRX) in the amyloid fibril is located on the surface and is accessible for the attachment of other proteins. In addition, the linker used is long enough to allow the free placement of TRX molecules around the fibril.

The next question is what state TRX is in the fibril. To determine TRX state we measured tryptophan fluorescence during amyloid formation. Note that there are no tryptophan residues in ABB, which means that in this way we observe only TRX. As we see, in the process of amyloid formation, the position of the peak of tryptophan fluorescence (I320/I380 intensity ratio) practically does not change (Figure 10b). In addition, we analyzed the stability of the thioredoxin structure. For this, denaturation by urea was investigated and compared free TRX, TRX as part of a fusion protein and TRX in mature amyloid fibrils. Structural changes were recorded by the change in the position of the peak of tryptophan fluorescence. The experiment showed a coincidence of denaturation curves (Figure 10c). This leads to two conclusions. First: thioredoxin maintains its stable spatial structure. Second: the interaction of TRX with ABB and its amyloid fibril is negligible. Otherwise, the stability of the TRX structure would be significantly changed. This fact opens up broad prospects for further usage of ABB and its variants as a platform for rational design of amyloid fibrils carrying functional domains.

## 4. Conclusions

Thus, we characterized ABB as an amyloidogenic protein and as a promising object for the amyloid structures engineering. The main stages of its amyloid formation were shown. In addition, ways to control this process were shown.

The results of detailed study focused on the kinetics of amyloid formation by ABB using the spectral and microscopic methods infer that this process involves four main stages.

(1) Non-fibrillar aggregates formation and accumulation take place at the first stage of the ABB amyloid fibril formation as evidenced by changes in the CD spectra of the specimen and changes in the morphology detected by electron microscopy. No binding of ThT amyloid specific dye is detected here. (2) The second stage is characterized by formation of the short fibrils with cross-beta structure as evidenced by the data on increasing intensity of ThT fluorescence. (3) Further changes in the amyloid morphology indicate that there is a third stage. According to the EM and AFM data, growth and accumulation of fibrils forming a dense network take place at this stage. (4) According to the fluorescence microscopy data, the fourth stage of amyloid formation by ABB can be differentiated, which involves the formation of large fibril bundles observed at long incubation times. The formation of the suprafibrillar structure is attributed to the fact that the backbone of an amyloid fibril typically contains only a small portion of the protein (~8–10 AAR) [1], while the remaining portion is somehow involved in the interaction between fibrils or in the formation of large bundles by them.

Undoubtedly, the protein engineering of amyloid structures and their use in practice requires knowledge of the properties of a current protein and methods for controlling the process of its amyloid formation. Such knowledge for ABB was the goal of our work. ABB, is one of many proteins for which the possibility of amyloid formation is shown, and at the same time one of the few that can be put into practice. For ABB, in our opinion, the most obvious application is to use it as a structural basis for functional amyloid fibrils. We have shown the possibility of this using TRX as an example. However, this requires verification and confirmation with other proteins or functional domains. A remarkable property of the ABB-based fusion protein is the possibility of its preparation and purification in monomeric water-soluble form and the subsequent controlled start of amyloid formation. To control the launch, as we have shown in this work, one can use a change in pH or disulfide bonds breaking. The rate of fibril formation can be accurately controlled by temperature and ionic strength.

In conclusion, we want to note the joyful fact that the albebetin, one of the first artificial proteins in the world, engineered under the leadership of O.B. Ptitsin, continues to evolve and find its new uses.

## Figures and Tables

**Figure 1 biomolecules-10-00241-f001:**
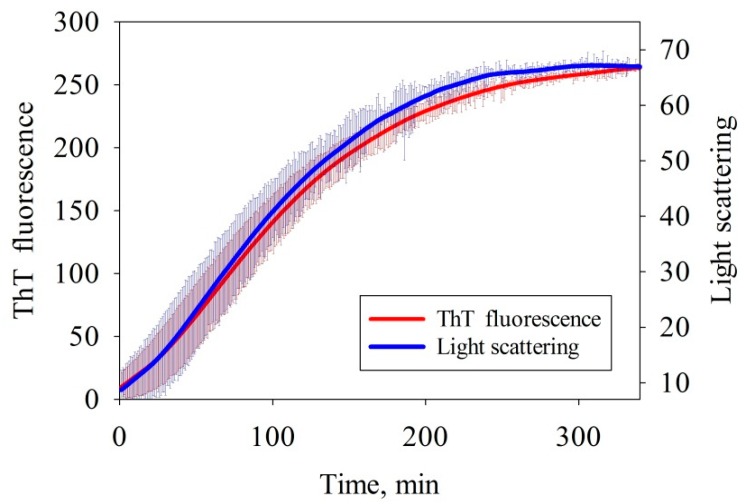
Time dependences of the intensity of thioflavin T (ThT) fluorescence and light scattering on amyloid formation by albebetin (ABB) in the 20 mM Tris-HCl buffer (pH 7.4) at 45 °C in the presence of 200 mM NaCl. The thickness of the solid lines corresponds to the instrumental errors; the hatching shows the dispersion during repeated repetition of the experiment.

**Figure 2 biomolecules-10-00241-f002:**
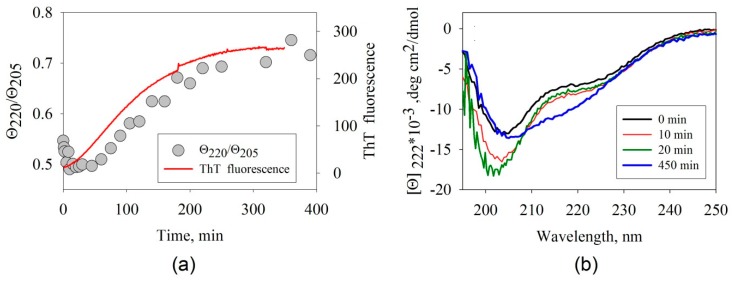
(**a**) Time dependences of the ratio of ellipticities, Θ_2205_/Θ_205_, and the intensity of ThT fluorescence recorded when studying the kinetics of amyloid formation by ABB. Instrumental errors correspond to the size of the characters and the thickness of the line. (**b**) The far-UV circular dichroism (CD) spectra of ABB measured before incubation (black), after 10-min incubation (red), 20-min incubation (green), and 450-min incubation (blue) at 45 °C in the 20 mM Tris-HCl buffer (pH 7.4) in the presence of 200 mM NaCl.

**Figure 3 biomolecules-10-00241-f003:**
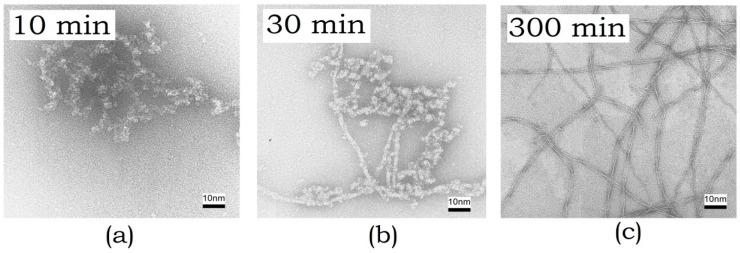
Electron microimages of amyloid structures formed by ABB recorded using negative staining with uranyl acetate after incubation at 45 °C for (**a**) 10 min, (**b**) 30 min, and (**c**) 300 min.

**Figure 4 biomolecules-10-00241-f004:**
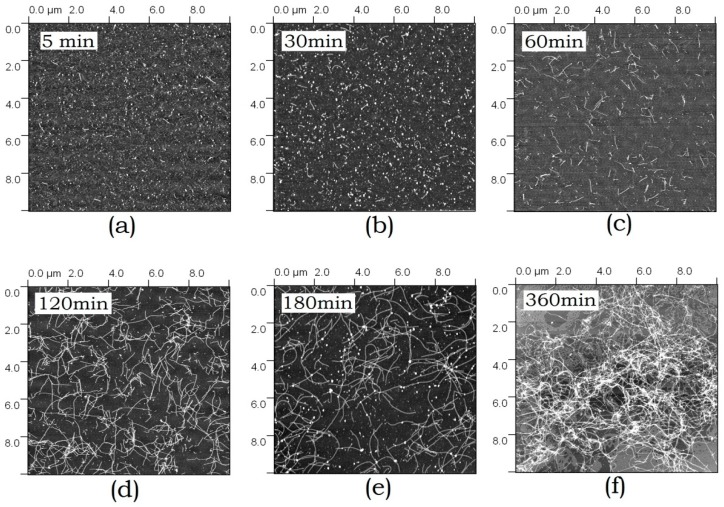
Atomic force microscopy (AFM) images of the amyloid structures formed by ABB upon 5 min-6 h (**a**–**f**) incubation in 20 mM Tris-HCl buffer (pH 7.4) in the presence of 200 mM NaCl at 45 °C.

**Figure 5 biomolecules-10-00241-f005:**
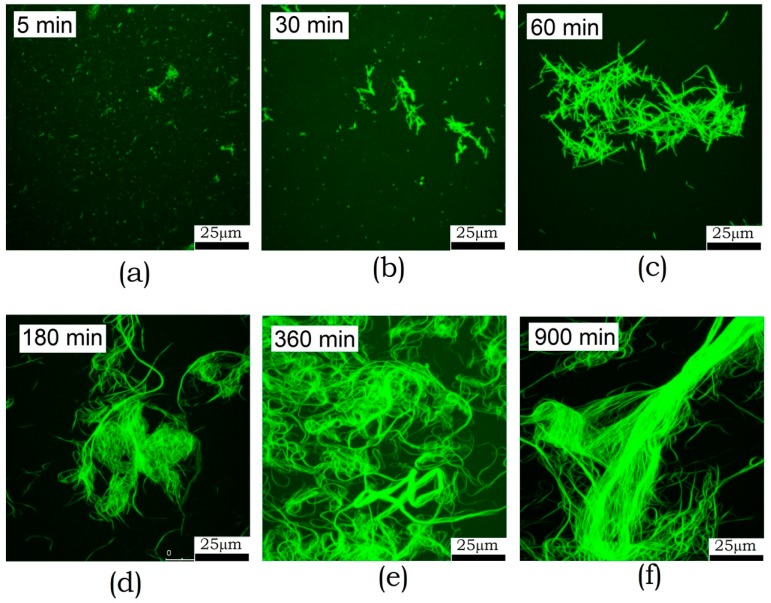
Fluorescence microscopy images of the amyloid structures formed by ABB upon 5 min-15 h (**a**–**f**) incubation in 20 mM Tris-HCl buffer (pH 7.4) containing 200 mM NaCl at 45 °C.

**Figure 6 biomolecules-10-00241-f006:**
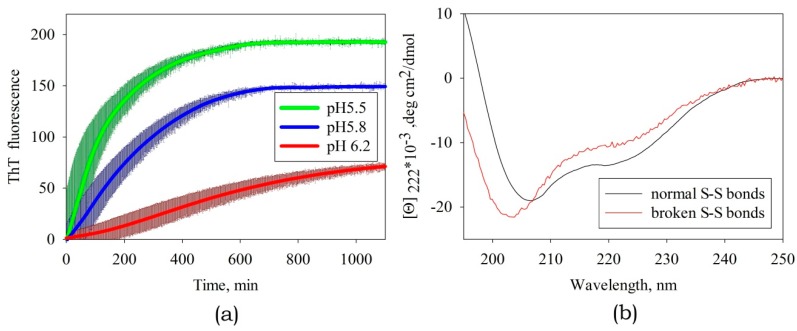
(**a**) The effect of pH on the rate of amyloid formation of ABB. The thickness of the solid lines corresponds to the instrumental errors; the hatching shows the dispersion during repeated repetition of the experiment. (**b**) The CD spectra of albebetin at pH 5.5 in the presence and in the absence of the disulfide bond reducing agent TCEP.

**Figure 7 biomolecules-10-00241-f007:**
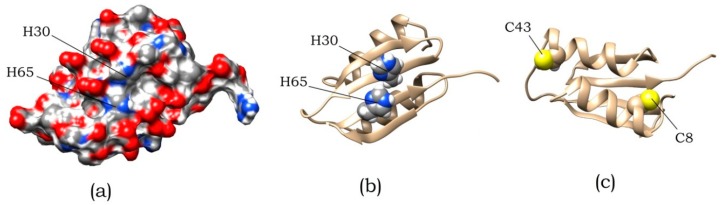
The spatial structure of ABB predicted using the online service robetta.org [30]. (**a**) Surface distribution of polar and charged atoms. (**b**) Location of residues H30 and H65. (**c**) Location of residues C8 and C43.

**Figure 8 biomolecules-10-00241-f008:**
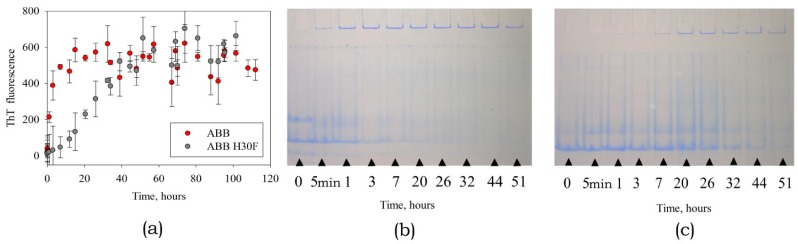
The kinetics of amyloid formation by the native ABB and its mutant variant ABB(H30F). (**a**) Time dependences of the intensity of ThT fluorescence. Analysis of the contents of monomeric and aggregated forms at different stages of incubation using gel electrophoresis in 9% PAGE gel under non-denaturing conditions for original ABB (**b**) and mutant variant ABB(H30F) (**c**).

**Figure 9 biomolecules-10-00241-f009:**
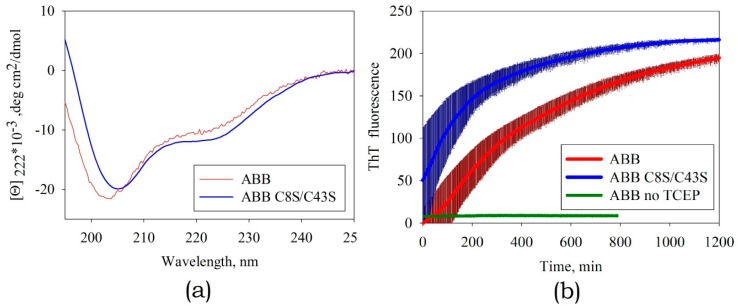
(**a**) The CD spectra of the mutant variant ABB(C8S/C43S) and original ABB in the presence of the reducing agent tris(2-carboxyethyl)phosphin hydrochloride (TCEP). (**b**) The kinetics of amyloid formation by the mutant variant ABB(C8S/C43S) and original ABB in the presence and absence of TCEP according to rising fluorescence of ThT. The thickness of the solid lines corresponds to the instrumental errors; the hatching shows the dispersion during repeated repetition of the experiment.

**Figure 10 biomolecules-10-00241-f010:**
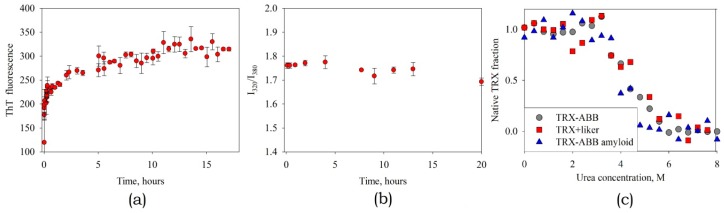
The kinetics of amyloid formation by the hybrid protein ABB-TRX observed by ThT fluorescence (**a**) and the ratio of the intensities of tryptophan fluorescence at 320 and 380 nm (**b**). TRX structure unfolding by urea for free TRX, TRX-ABB fusion and TRX-ABB fusion in the amyloid state (**c**).

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
