# Peer review of "The Kinetics of Amyloid Fibril Formation by de Novo Protein Albebetin and Its Mutant Variants"

_biomolecules, 2020, doi:10.3390/biom10020241_

Round 1

Reviewer 1 Report

In the present manuscript the authors use a series of microscopic and spectral studies to draw a phenomenological picture of the aggregation mechanism of the de novo deigned protein albebetin. Based on a qualitative integrated analysis the authors propose four stages for the aggregation mechanism: (1) non-fibrillar aggregates, (2) short fibrils with cross beta structure , (3) formation of a dense fibril network, and finally (4) formation of large fibril bundles. The authors further explore the impact of the oxidation state of the disulphide bond on the aggregation potential, and investigate the impact of selected single point mutations located at the termini on aggregation. In my view there is nothing wrong with the manuscript but to be honest there is also nothing very exciting about the reported data.  In particular, the manuscript could become potentially more interesting if the authors could put forward some mechanistic rationale for the effects of mutations (is there an increase in SASA upon mutation, repacking of the hydrophobic core, increased mobility of some structural region of the protein, how does the Tm changes upon mutation? Is there any loss of stability? How does the charge distribution change? etc) on the aggregation process. Regarding the disulphide bond the authors mention that “The results of

our study may indicate that the conformational mobility of the polypeptide chain is very important for efficient amyloid formation and the disulfide bond plays a crucial role as one of the factors stabilizing the non-amyloid form of protein.” . How did the authors got to this conclusion? I don’t see any quantitative analysis that allows one to conclude as such.  Overall, as it stands the discussion is too descriptive and speculative. Also, the following minor points should be taken into account in the revised version of the manuscript:

 -There are some sentences that need to be revised in terms of English usage. Examples: “There exist proteins that 
exhibit functional activity only when having the amyloid configuration”; “The formation of amyloid structures is typically mediated by protein transformation due to certain factors to an 
intermediate state whose properties are similar to those of a molten globule”;  “repeated repetition of the experiment. “; “Although it does not see oligomers at all”; “Such external factors as pH and the presence of agents breaking disulfide bonds were found to be most significant. ”

- The authors should add citations to support the following sentences:

(1) ʉ۬It was demonstrated in a number of studies that such factors as pH of the medium, disulfide bonds or point mutations affect the kinetics of amyloid formation.

(2) The formation of 
amyloid structures is typically mediated by protein transformation due to certain factors to an intermediate state whose properties are similar to those of a molten globule (the presence of a 
secondary structure but no dense packing of side groups, as well as the high conformational 
mobility of a molecule). In this case, in particular, the following articles should be considered:

https://doi.org/10.1126/science.1214203

https://doi.org/10.1016/j.jmb.2012.06.020

Section 3.3, “Constructing and synthesizing mutant variants of albebetin”, should be moved to Materials and Methods The sentence “The mutations trigger many severe diseases collectively known as 
amyloidosis. In this connection, it would be interesting to model such amino acid substitutions in the ABB that could change the kinetics of amyloid formation in either side.” 
, does not make sense. It seems that amyloidosis are diseases caused exclusively by mutations, which is not true.

Author Response

Response to Reviewer 1

Dear reviewer. Thank you for carefully reading our manuscript and valuable comments. Any criticism that makes us better is important to us. With your permission, we will give an answer to your review step by step.

In the present manuscript the authors use a series of microscopic and spectral studies to draw a phenomenological picture of the aggregation mechanism of the de novo deigned protein albebetin. Based on a qualitative integrated analysis the authors propose four stages for the aggregation mechanism: (1) non-fibrillar aggregates, (2) short fibrils with cross beta structure , (3) formation of a dense fibril network, and finally (4) formation of large fibril bundles. The authors further explore the impact of the oxidation state of the disulphide bond on the aggregation potential, and investigate the impact of selected single point mutations located at the termini on aggregation. In my view there is nothing wrong with the manuscript but to be honest there is also nothing very exciting about the reported data.

This article is written for the release in a journal issue dedicated to O.B. Ptitsin. Albebetin - one of the first artificial proteins in the world was designed and created under his leadership. The article a reflected the modern research of this protein at the threshold of its step into the engineering of amyloid structures. Yes, Albebetine is only one of many amyloidogenic proteins, but it is undoubtedly one of the few whose use, in our opinion, is possible in practice.

In particular, the manuscript could become potentially more interesting if the authors could put forward some mechanistic rationale for the effects of mutations (is there an increase in SASA upon mutation, repacking of the hydrophobic core, increased mobility of some structural region of the protein, how does the Tm changes upon mutation? Is there any loss of stability? How does the charge distribution change? etc) on the aggregation process.

Albebetin does not have a solid native state and is in a molten globule state. Structural and thermodynamic studies for such proteins are very difficult. At the moment, we have only theoretically predicted structures at our disposal. Based on them, it is possible to make the proposed calculations only with large errors. The practical value of such calculations is very low. The most reliable one is the analysis of the surface distribution of polar and charged residues. Such a procedure was carried out by us earlier and is now added to the article.

Regarding the disulphide bond the authors mention that “The results of our study may indicate that the conformational mobility of the polypeptide chain is very important for efficient amyloid formation and the disulfide bond plays a crucial role as one of the factors stabilizing the non-amyloid form of protein.” . How did the authors got to this conclusion? I don’t see any quantitative analysis that allows one to conclude as such.  Overall, as it stands the discussion is too descriptive and speculative.

The formation of disulfide bonds according to the results of our experiments unambiguously prevents the formation of amyloid fibrils by free Albebetine and its mutant forms, as well as in a fusion protein. This can be explained by two variants: the first is the restriction of the mobility of the polypeptide chain, the second is the participation of cysteine residues in the formation of the amyloid fibril core in the restored state. Predicting the structure of a protein with an intermolecular disulfide bond gives a structure significantly different from that predicted for a protein without a SS bond. We give this structure for you but not for the article. At the same time, these structures have a similar distribution and content of the secondary structure, which creates difficulties for diversity them by spectral methods.

Also, the following minor points should be taken into account in the revised version of the manuscript:

 -There are some sentences that need to be revised in terms of English usage. Examples: “There exist proteins that 
exhibit functional activity only when having the amyloid configuration”; “The formation of amyloid structures is typically mediated by protein transformation due to certain factors to an 
intermediate state whose properties are similar to those of a molten globule”;  “repeated repetition of the experiment. “; “Although it does not see oligomers at all”; “Such external factors as pH and the presence of agents breaking disulfide bonds were found to be most significant. ”

Thanks for the comments. The text of the manuscript was corrected.

- The authors should add citations to support the following sentences:

(1) ʉ۬It was demonstrated in a number of studies that such factors as pH of the medium, disulfide bonds or point mutations affect the kinetics of amyloid formation.

This phrase is absent in the current edition of the article, but it is supported by research in articles 31, 32 and 34.

(2) The formation of 
amyloid structures is typically mediated by protein transformation due to certain factors to an intermediate state whose properties are similar to those of a molten globule (the presence of a 
secondary structure but no dense packing of side groups, as well as the high conformational 
mobility of a molecule). In this case, in particular, the following articles should be considered:

https://doi.org/10.1126/science.1214203

https://doi.org/10.1016/j.jmb.2012.06.020

We added a link to article by H. Krobath et al. and some discussion of the mechanism for increasing the aggregation ability of intermediate conformational states of proteins. 

Section 3.3, “Constructing and synthesizing mutant variants of albebetin”, should be moved to Materials and Methods

Thanks for the comment, we have moved this part to the appropriate section (“Materials and Methods”).

The sentence “The mutations trigger many severe diseases collectively known as 
amyloidosis. In this connection, it would be interesting to model such amino acid substitutions in the ABB that could change the kinetics of amyloid formation in either side.” 
, does not make sense. It seems that amyloidosis are diseases caused exclusively by mutations, which is not true.

We agree, the phrase is not entirely correct. Corrected.

Kind regards,

Vitalii Balobanov,

Reviewer 2 Report

The manuscript aims to study the kinetic pathway of amyloid fibril formation by de novo protein albebetin. The authors utilized some routinely used biophysical spectroscopy and microscopy assays to probe for the amyloidogenesis of albebetin. Some point mutations that alter the kinetics of fibrillation are also studied. Before I discuss the scientific merit of the current work, let me briefly discuss the previous work on the artificial protein albebetin's amyloid formation by the same author(s). The protein's structural and biophysical behavior was first reported more than two decades ago. The amyloid process has been probed in several papers in the last decade, including the potential cytotoxicity of abebetin oligomers containing amyloid-like structures (two papers in 2004 and 2006). Therefore, I am not sure if the current study is entirely novel.

In the current paper, the authors claim that they observed multiple stages of amyloid formation and their dependence on the di-sulfide bond perturbing mutations and external solution conditions. These are routine studies of proteins that form amyloids and they do not provide any new information that could be of substantial interest. The CD results if Fig. 2 are confusing and I could not find any proper explanation of why there is an initial decay of 220/205 signal.  I am also not sure what new information confocal microscopy can provide on top of EM and AFM. Furthermore, explanations and quantification of why covalent binding of a thioredoxin molecule to ABB does not make ABB lose its amyloidogenic properties is not present in the manuscript. The interpretation that there are four stages in ABB amyloidogenesis is not supported by convincing experimental data. Finally, Just like the experimental section, the conclusion reads like a list of things rather than providing a comprehensive mechanistic picture. 

Author Response

Response to Reviewer 2

Dear reviewer. Thank you for carefully reading our manuscript and valuable comments. Any criticism that makes us better is important to us. With your permission, we will give an answer to your review step by step.

The manuscript aims to study the kinetic pathway of amyloid fibril formation by de novo protein albebetin. The authors utilized some routinely used biophysical spectroscopy and microscopy assays to probe for the amyloidogenesis of albebetin. Some point mutations that alter the kinetics of fibrillation are also studied. Before I discuss the scientific merit of the current work, let me briefly discuss the previous work on the artificial protein albebetin's amyloid formation by the same author(s). The protein's structural and biophysical behavior was first reported more than two decades ago. The amyloid process has been probed in several papers in the last decade, including the potential cytotoxicity of abebetin oligomers containing amyloid-like structures (two papers in 2004 and 2006). Therefore, I am not sure if the current study is entirely novel.

Indeed, this is not the first study of the amyloidogenic properties of ABB. However, the mentioned articles are devoted to the study of amyloidogenic properties and cytotoxicity of fibrils of biologically active variants of ABB protein carrying functional peptides of factor HLDF and Interferon-alpha, in comparison with ABB. The studies were carried out in other conditions and in slightly different aspects. The experimental data in current article is completely new. In the present work, we applied an approach in which we measured by all methods in one experiment. This allowed us to correlate the kinetic data obtained by different methods and solve the problems of poor kinetic data reproducibility. It should be noted that, due to the nucleation-elongation nature of amyloid aggregation, the scatter of kinetic data during repetition of the experiment exceeds the difference between the methods in one experiment. Our approach allowed us to remove this restriction and obtain the presented result. In addition, the results of the study of amyloid aggregation of three mutant variants ABB (H30F), ABB (H65F), ABB (C8S / C43S) and the fusion protein TRX-ABB are presented, which is also new data.

In the current paper, the authors claim that they observed multiple stages of amyloid formation and their dependence on the di-sulfide bond perturbing mutations and external solution conditions. These are routine studies of proteins that form amyloids and they do not provide any new information that could be of substantial interest.

 This article is written for the release in a journal issue dedicated to O.B. Ptitsin. Albebetin - one of the first artificial proteins in the world was designed and created under his leadership. The article a reflected the modern research of this protein at the threshold of its step into the engineering of amyloid structures. Yes, Albebetine is only one of many amyloidogenic proteins, but it is undoubtedly one of the few whose use, in our opinion, is possible in practice.

The CD results if Fig. 2 are confusing and I could not find any proper explanation of why there is an initial decay of 220/205 signal.

Perhaps we have not given the most characteristic spectrum. We added spectrum for a point of 20 min, see Figure 2. (b). The spectra show an increase in ellipticity by 205, which indicates an increase in the fraction of the expanded structure during the formation of primary aggregates.

I am also not sure what new information confocal microscopy can provide on top of EM and AFM.

Fluorescence microscopy has a lower resolution compared to AFM and electron microscopy. Although it does not see oligomers at all, it allows one to examine larger fields and larger structures. This may be important for the design and practical application of functional amyloid fibrils. Thus, we found this method adequate to study the suprafibrillar structure of amyloids.

 Furthermore, explanations and quantification of why covalent binding of a thioredoxin molecule to ABB does not make ABB lose its amyloidogenic properties is not present in the manuscript.

In our work we used pET-32 Xa/LIC vector for high-level expression of polypeptides fused with the 109 AA thioredoxin (TRX) protein. The amino acid sequences ABB and TRX are connected through a rather long (about 5-6 kDa) linker region so that TRX is located at the N-terminus of the ABB molecule. We have shown that the ABB-TRX fusion molecule has the ability to form amyloid fibrils Figure 10 (a). We analyzed the stability of the thioredoxin structure to demonstrate the maintenance of the TRX native structure, Figure 10 (b) and – new - (c). That data obtained allow us to conclude that the ABB molecule in the composition of the fusion protein has amyloidogenic properties, and the TRX molecule retains its native structure. This fact can be explained by the assumption that the N-terminus of albetine in the amyloid structure is located on the surface of the fibril, which allows the TRX molecule (or any other domain) to be exposed to the solution and not interfere with the formation/growth of the fibril. This is facilitated by the presence of a long linker between the ABB and TRX polypeptide chains, which, in addition, removes the steric difficulties in packing large TRX molecules near the fibril. A description of this has been added to the article.

The interpretation that there are four stages in ABB amyloidogenesis is not supported by convincing experimental data.

The combined analysis of experimental kinetic curves shows the presence of at least four stages. These stages are reliably distinguished in the study by many methods in one experiment. Therefore we suppose that four stages of the formation of ABB amyloid fibrils can be distinguished, since using spectral and microscopic methods during incubation over different time periods, we observed four types of amyloid formations, different in morphology: 1.non-fibrillar aggregates, 2.short fibrils, 3. fibrillar network, 4. suprafibrillar structure. Thus, this hypothesis based on our experimental data seems reasonable to us.

Finally, Just like the experimental section, the conclusion reads like a list of things rather than providing a comprehensive mechanistic picture.

Thank your for you comment, we revised conclusion of the manuscript.

Reviewer 3 Report

In the manuscript “The kinetics of amyloid fibril formation by de novo protein albebetin and its mutant variants”, the Authors characterized the kinetics of amyloid aggregation of the albebetin variants by standard approaches (ThT fluorescence, circular dichroism, electron and atomic force microscopies, and confocal fluorescence microscopy). The manuscript follows the typical workflow of a very large number of publications devoted to the study of amyloid aggregation of specific proteins.

The experiments appear well designed and the main conclusions are supported by the reported data.

In my opinion, the manuscript should be further improved to clarify some points.

Major point.

1) Lines 394-399. In my opinion, the supposed native-like structure of the TRX domain represents the most interesting portion of this manuscript. However, the intrinsic fluorescence emission is not sufficient to demonstrate the maintenance of the TRX native structure and function in the fibrils. I suggest employing other domains whose function/structure can be unequivocally tested in the fusion protein such as the GFP (by fluorescence) or enzymes (by enzymatic assays).   

Minor points

1) Lines 119-121. Please, specify how the extinction coefficient was determined.

2) Lines 122-129. Please, specify the final concentration of ThT.

3) Line 126. “the images were recorded at 480 nm ” should be “the emission was recorded at 480 nm”

4)

-Lines 153-154. “After incubation, the protein was diluted with Prolong Gold anti-fade solution (Invitrogen) at a 1:1 ratio.”

-Lines 194-196. …….

-Lines 265-266. “The images of all layers are merged together.”

-Caption of figure 8 b and 8 c.

These statements require more details, clarifications or proper references.  

5) -Line 168. Why “arbitrary”? 

-Line 308. Why “ambiguous”? Perhaps “protein-specific”?

-Several times (for instance line 350): original type, native ABB, or wild type?

6) Lines 333-339. Have the DNA constructs been sequenced to confirm the wanted mutations?

7) Figure 10. Please, provide the emission spectra of the intrinsic fluorescence.

8) Line 138. It is not clear to me if the Authors calculated the protein secondary structure from circular dichroism using the Provencher–Glöckner method or if they only changed the CD unit from that provided by the instrument to Molar ellipticity.

Author Response

Response to Reviewer 3

Dear reviewer. Thank you for carefully reading our manuscript and valuable comments. Any criticism that makes us better is important to us. With your permission, we will give an answer to your review step by step.

In the manuscript “The kinetics of amyloid fibril formation by de novo protein albebetin and its mutant variants”, the Authors characterized the kinetics of amyloid aggregation of the albebetin variants by standard approaches (ThT fluorescence, circular dichroism, electron and atomic force microscopies, and confocal fluorescence microscopy). The manuscript follows the typical workflow of a very large number of publications devoted to the study of amyloid aggregation of specific proteins.

This article is written for the release in a journal issue dedicated to O.B. Ptitsin. Albebetin - one of the first artificial proteins in the world was designed and created under his leadership. The article a reflected the modern research of this protein at the threshold of its step into the engineering of amyloid structures. Yes, Albebetine is only one of many amyloidogenic proteins, but it is undoubtedly one of the few whose use, in our opinion, is possible in practice.

In my opinion, the manuscript should be further improved to clarify some points.

Major point.

Lines 394-399. In my opinion, the supposed native-like structure of the TRX domain represents the most interesting portion of this manuscript. However, the intrinsic fluorescence emission is not sufficient to demonstrate the maintenance of the TRX native structure and function in the fibrils. I suggest employing other domains whose function/structure can be unequivocally tested in the fusion protein such as the GFP (by fluorescence) or enzymes (by enzymatic assays).

We have expanded the description of this part of the work. In addition, we added a description of the experiment to determine the stability of thioredoxin structure in the free state and in the amyloid fibrils. We included these results into the manuscript, please, see Figure 10 (c) and its description. I agree with your opinion that it is necessary to verify the detected effects on other proteins and domains. Work in this direction will be the next step in our research.

Minor points

Lines 119-121. Please, specify how the extinction coefficient was determined.

This is a theoretically calculated extinction coefficient. Revised.

Lines 122-129. Please, specify the final concentration of ThT.

The final concentration of Thioflavin T was 0.1 mkM. Revised.

Line 126. “the images were recorded at 480 nm ” should be “the emission was recorded at 480 nm”

Revised.

4)

-Lines 153-154. “After incubation, the protein was diluted with Prolong Gold anti-fade solution (Invitrogen) at a 1:1 ratio.”

-Lines 194-196. …….

-Lines 265-266. “The images of all layers are merged together.”

-Caption of figure 8 b and 8 c.

These statements require more details, clarifications or proper references.

Revised.

-Line 168. Why “arbitrary”? 

Changed to «stochastic».

-Line 308. Why “ambiguous”? Perhaps “protein-specific”?

Yes, it’s will be better.  Changed.

-Several times (for instance line 350): original type, native ABB, or wild type?

 We left the option "original". Revised.

Lines 333-339. Have the DNA constructs been sequenced to confirm the wanted mutations?

We included this information into the “Methods” section of the manuscript.

Figure 10. Please, provide the emission spectra of the intrinsic fluorescence.

We do not believe that the protein spectrum will provide additional information in the context of this experiment. For equilibrium unfolding, the spectra of native and denatured protein are given here. The values I320 / I380 for native and denatured were 1.73 and 0.8, respectively.

Line 138. It is not clear to me if the Authors calculated the protein secondary structure from circular dichroism using the Provencher–Glöckner method or if they only changed the CD unit from that provided by the instrument to Molar ellipticity.

The graphs show molar ellipticity calculated taking into account the concentration of the protein, its amino acid composition and the size of the cuvette. The calculation of the secondary structure content for alpha + beta proteins by the Provencher – Glöckner method is ambiguous. Therefore, in order not to mislead readers, we do not provide these figures in the article. Revised.

Kind regards,

Vitalii Balobanov,

Round 2

Reviewer 1 Report

The authors have substantially reviewed the manuscript; their responses to my criticisms are satisfactory. I recommend publication once the typos have been removed (eg, line 38 -areproteins; line 45 - supouse; line 148 - anaisis; line 318 - during repeated repetition...etc) 

Author Response

Dear reviewer.

Thank you for your careful manuscript reading and valuable comments. We took your comments into account and made changes in the manuscript.

Kind regards,

Vitalii Balobanov,

Corresponding Author.

Reviewer 3 Report

In my opinion, this paper has improved significantly from its previous submission.

Some typos or units (line 160: is mkM an accepted unit for concentration?) need to be corrected.   

Author Response

Dear reviewer.

Thank you for your careful manuscript reading and valuable comments. We took your comments into account and made changes in the manuscript.

Kind regards,

Vitalii Balobanov,
